# Genomic Alterations, Gene Expression Profiles and Functional Enrichment of Normal-Karyotype Acute Myeloid Leukaemia Based on Targeted Next-Generation Sequencing

**DOI:** 10.3390/cancers15051386

**Published:** 2023-02-22

**Authors:** Angeli Ambayya, Rozaimi Razali, Sarina Sulong, Ezzanie Suffya Zulkefli, Yee Yee Yap, Jameela Sathar, Rosline Hassan

**Affiliations:** 1Department of Haematology, School of Medical Sciences, Health Campus, Universiti Sains Malaysia, Kubang Kerian 16150, Malaysia; 2Clinical Haematology Referral Laboratory, Haematology Department, Hospital Ampang, Ministry of Health Malaysia, Ampang 68000, Malaysia; 3Department of Biomedical Science, College of Health Sciences, QU-Health, Qatar University, Doha P.O. Box 2713, Qatar; 4Human Genome Centre, School of Medical Sciences, Health Campus, Universiti Sains Malaysia, Kubang Kerian 16150, Malaysia; 5Hospital Universiti Sains Malaysia, Health Campus, Universiti Sains Malaysia, Kubang Kerian 16150, Malaysia

**Keywords:** acute myeloid leukaemia, next generation sequencing, gene expression, somatic variants, functional enrichment

## Abstract

**Simple Summary:**

The characterisation of normal-karyotype acute myeloid leukaemia (AML-NK) requires further refinement based on genetic variations. In this study, we ascertained genomic biomarkers via DNA and RNA sequencing in eight AML-NK patients during diagnosis and after achieving complete remission. We discovered putative variants affecting gene regulation and functional enrichments dysregulating transcription and DNA-binding transcription activator activity RNA polymerase II-specific in our cohort.

**Abstract:**

Characterising genomic variants is paramount in understanding the pathogenesis and heterogeneity of normal-karyotype acute myeloid leukaemia (AML-NK). In this study, clinically significant genomic biomarkers were ascertained using targeted DNA sequencing and RNA sequencing on eight AML-NK patients’ samples collected at disease presentation and after complete remission. In silico and Sanger sequencing validations were performed to validate variants of interest, and they were followed by the performance of functional and pathway enrichment analyses for overrepresentation analysis of genes with somatic variants. Somatic variants involving 26 genes were identified and classified as follows: 18/42 (42.9%) as pathogenic, 4/42 (9.5%) as likely pathogenic, 4/42 (9.5%) as variants of unknown significance, 7/42 (16.7%) as likely benign and 9/42 (21.4%) as benign. Nine novel somatic variants were discovered, of which three were likely pathogenic, in the CEBPA gene with significant association with its upregulation. Transcription misregulation in cancer tops the affected pathways involving upstream genes (CEBPA and RUNX1) that were deregulated in most patients during disease presentation and were closely related to the most enriched molecular function gene ontology category, DNA-binding transcription activator activity RNA polymerase II-specific (GO:0001228). In summary, this study elucidated putative variants and their gene expression profiles along with functional and pathway enrichment in AML-NK patients.

## 1. Introduction

Acute myeloid leukaemia (AML) is a clonal haematopoietic stem cell disorder characterised by genetic heterogeneity with diverse clinical outcomes in patients. For decades, cytogenetics has been crucial in diagnosing AML, as it provides a snapshot of genome-wide structural and copy number changes. Established recurrent structural variations such as t(8;21), t(15;17), inv(16) and del(5/7) are pivotal in the diagnosis and prognosis of AML, as these abnormalities partake in leukaemogenesis [1]. However, almost half of all adult AML cases are diagnosed with a normal karyotype (NK) without structural aberrations identified during cytogenetic analysis. Varying frequencies of normal karyotypes were reported by different countries, and those frequencies ranged between 25% to 70% [2,3]. Although the European Leukaemia Network (ELN) in 2017 and 2022 classified AML-NK as an intermediate prognosis, the clinical outcomes of the patients in this group are diverse and arduous to define [4,5,6,7]. In recent years, the advent of novel and more sensitive platforms such as next-generation sequencing enabled the interrogation of the AML-NK genome to detect cryptic and subtle genomic aberrations that are otherwise undetectable with cytogenetics analysis. These genomic aberrations were indicated to be useful, and several findings were applied to the National Comprehensive Cancer Network (NCCN) guidelines [8].

Some of the genomic aberrations that are included in the diagnosis and prognosis of AML-NK are FMS-like tyrosine kinase 3 internal tandem duplication (FLT3-ITD) mutations that confer poor prognoses [6,9,10] if detected without nucleophosmin (NPM1) [6,11,12] concomitant mutations. NPM1 mutations without FLT3-ITD mutations are associated with good prognoses, and CCAAT enhancer-binding protein alpha (CEBPA) mutations also indicate a good prognosis if biallelic mutation [6,11,12,13,14] is present. This is also true of other myeloid gene-related mutations. Although some genomic aberrations have been established as diagnostic and prognostic markers, some genetic aberrations, such as Runt-related transcription factor 1 (RUNX1) [12] and Tet methylcytosine dioxygenase 2 (TET2) [15,16], are still controversial, as their role in clinical studies is yet to be fully understood.

On the other hand, gene expression profiling improves the understanding of the AMK-NK subgroup and further refines risk stratification and clinical outcomes. Several studies elucidated the distinct gene expression profiles between the AML-NK patients and refinement based on genomic aberrations such as the mutational status of prognostic genes (FLT3-ITD, CEBPA and NPM1) [17,18,19,20]. The study of differentially expressed genes (DEG) profiles provides insights into the perturbed pathways associated with genomic markers in AML-NK. The discovery of activated and inactivated pathways enables prediction of a patient’s response to specific targeted therapies. For example, the perturbed RAS signalling pathway inspired the development of RAS-targeted therapy in AML patients [21].

Although considerable progress has been made in discovering genomic biomarkers in AML-NK, elucidating the genetic heterogeneity of this disorder, the backbone of induction chemotherapy with cytarabine (Ara-C) and anthracyclines has not changed in the last three decades. Monitoring patients’ responses to chemotherapy was limited in AML-NK patients due to the paucity of suitable markers for minimal residual disease (MRD) assessment in the past. Nevertheless, now, NGS technologies enable deciphering the clonal heterogeneity of the AML-NK genome to identify specific genomic biomarkers suitable for MRD monitoring [22,23]. The persistence of the genomic biomarkers of MRD at the time of complete remission (CR) serves as an independent prognosis for survival, as reported by Jongen-Laverencic et al. They analysed 482 AML patients using a targeted NGS panel at two time points: at presentation and CR after induction chemotherapy. In that study, the authors reported that about 50% of the mutations detected at presentation persisted after CR and that most of these mutations conferred increased relapse risk [24,25]. Other studies suggested that NGS-based MRD monitoring has better sensitivity and specificity than conventional MRD monitoring techniques such as morphological review and flow cytometry immunophenotyping [26,27].

To our knowledge, no comprehensive study on AML-NK integrating DNA and RNA sequencing with functional enrichment has been published. Therefore, we explored the genomic aberrations in the AML-NK genome using a targeted NGS panel and RNA sequencing for gene expression profiling to disclose DEGs. Next, we elucidated functional and pathway enrichment by performing overrepresentation analysis (ORA) in AML-NK patients affected by genomic aberrations and DEGs. Subsequently, we identified genomic biomarkers that could be useful in MRD monitoring for AML-NK patients.

## 2. Materials and Methods

### 2.1. Patient Samples and Ethics Statement

A cohort of eight matched de novo AML-NK patient samples at two time points, presentation (DX1-8) and after remission attainment (RX1-8) following completion of consolidation chemotherapy, were included in this study. All patients were diagnosed based on the latest WHO [12] criteria, and their responses to treatment were classified based on the International Working Group Criteria [28,29]. Patients were included if their cytogenetic findings were normal at presentation using the standard G-banding karyotype analysis of 20 metaphase chromosomal spreads obtained from bone marrow aspirates. Multiplex RT-qPCR was performed using a Leukaemia Q-Fusion Screening Kit (QuanDx, San Jose, CA, USA) for simultaneous detection of 30 fusion genes to rule out chromosomal aberrations. All samples were screened for prognostic markers using qualitative PCR and high-resolution melting analysis to detect FLT3-ITD and NPM1 mutations. The remission statuses of patients at CR1 were assessed based on post-induction chemotherapy that included a combination of cytarabine and anthracycline (daunorubicin) and consolidation chemotherapy that included high-dose cytarabine (HIDAC), a combination of mitoxantrone and cytarabine (MIDAC) or a combination of fludarabine, high-dose cytarabine and granulocyte colony-stimulating factor (FLAG). Patients’ remission at CR2 was assessed after salvage therapy using a combination of fludarabine, cytarabine, granulocyte-colony stimulating factor and idarubicin (FLAG-IDA). MRD monitoring was based on a morphological leukaemia-free state (bone marrow blast count of less than 5%) and either negativity of genetic markers (NPM1 and/or FLT3-ITD) or multiparametric flow cytometry for cases without genetic markers (absence of a leukaemia-associated immunophenotype) [6]. These samples were retrospectively selected from the Clinical Haematology Referral Laboratory, Hospital Ampang. Ethical approvals were obtained from the Medical Research Ethics Committee of the Ministry of Health Malaysia (NMRR 17-1929-36614) and Universiti Sains Malaysia (USM/JEPeM/21010107).

### 2.2. Targeted DNA Sequencing Using Archer Dx

Patients’ gDNA was extracted from blood collected in K2-EDTA using a Qiamp DNA Blood Minit Kit (Qiagen, Hilden, Germany) according to the manufacturer’s protocol. Genomic DNA concentration and purity were assessed by using a NanoDrop ND-1000 UV-VIS Spectrophotometer. Library preparation was performed using Archer HGC and VariantPlex Myeloid (Boulder, CO, USA) (Appendix A). The following Illumina’s adapter sequences were used for library preparation: read 1 (AGATCGGAAGAGCACACGTCTGAACTCCAGTCA) and read 2 (AGATCGGAAGAGCGTCGTGTAGGGAAAGAGTGT). Libraries were then quantified and normalised using qPCR (Kapa Biosystems, Wilmington, MA, USA). Sequencing was performed by Theragen Etex Inc. (Seoul, Korea) using Novaseq 6000, 150 bp paired-end analysis with a minimum 7M read per sample (presentation sample) and 30M (CR1/CR2 sample). Data analyses that included read quality cleaning, error correction, genome alignment and variant detection and annotation were performed on the ArcherDx platform (version 6.2) (Appendix A). The settings were optimised to include coding and non-coding sequences with default read filtering criteria, including a cut-off of allele frequency (AF) ≥ 0.027 and gnomAD AF ≤ 0.05. Raw sequencing data in Fastq format were used as the input file, and reads were mapped to the Hg 19 build of the human genome. The variant calling was conducted using three algorithms: Vision, Freebayes and LoFreq tools. A vision variant caller was used to call the somatic variants confined within the targeted mutation genes (Appendix A). The Freebayes variant caller utilised somatic variants, whereas Lofreq was used to detect somatic variants with AF frequencies. Variant annotation was done using the ArcherDx platform and verified using the Variant Effect Predictor tool (Ensembl) [30]. The variant allele frequency (VAF) was used to deduce the origin of a variant: germline or somatic. A variant was considered to be of germline origin if the VAF was between 50% to 100%, and it was then excluded from the analysis to ensure only somatic variants were included in this study [31].

All annotated variants were reviewed manually, and a series of filtering and prioritising was applied based on the following criteria: (1) variant consequences as defined by Sequence Ontology (SO) [32] based on Ensembl’s variation calculation filtering to include high-impact and selected moderate-impact variants (i.e., missense and protein-altering variants) [33], (2) exclusion of variants with minor allele frequencies (MAFs) of ≥ 1% in healthy populations’ databases (1000 Genomes Project, ExAC, gnomAD, and ESP6500) and (3) exclusion of likely sequencing errors (variants with VAF < 5%) [34]. Next, filtered variants were manually inspected using Integrated Genome Viewer (IGV) software (Broad Institute, Cambridge, MA, USA) and scrutinised against Varsome [35], dbSNP [36], Clinvar [37] and COSMIC [38] to assess previous reports on pathogenicity. Novel putative variants that were not reported were then evaluated based on computational variant effect prediction tools for coding regions (SIFT [39], Provean [40], Mutation Assessor [41], FATHMM [42], LRT [43]) and non-coding regions (CADD [44]). All variants were classified based on the American College of Medical Genetics (ACMG)’s recommendations and those of the Genomics and the Association for Molecular Pathology (AMP) and the American Society of Clinical Oncology [45,46]. Variants were categorised into five main groups: (1) pathogenic, (2) likely pathogenic, (3) uncertain significance, (4) likely benign and (5) benign. The ArcherDx platform provided AMP guideline-specified somatic categories as Tier I to IV, with the first tier indicating strong clinical significance and the fourth tier deemed benign/likely benign [45].

### 2.3. Sanger Sequencing

Sanger sequencing was performed on a recurrent ASXL1 c.1934dup G646WfsTer12 variant detected in 75% of the samples to verify if the variants with VAF < 5% were genuinely sequencing errors in this study (Appendix A: ASXL1 NM_015338.5:c.1934dup (VAF < 5%) validation, Appendix A). The IDT Primer Quest (Integrated DNA Technologies, Inc. was used to design the primers for the variant of interest (ASXL1c.1934dup) and checked for specificity using the NCBI Primer-BLAST. Optimisation of the annealing temperature DNA input was performed, followed by mastermix preparation and PCR cycles. The PCR thermocycling protocols were as follows: 98 °C for 10 s for initial denaturation, 60 °C for 5 s and 72 °C for 15 s (30 cycles), followed by a final extension at 72 °C for 1 min. The PCR products were visualised using Agilent’s D1000 Screentape with Agilent’s 2200 TapeStation system. PCR product purification was performed using Geneall Exspin PCR SV (103-102) according to the manufacturer’s instructions, and the samples were assessed for purity using the Nanodrop spectrophotometer. Samples with acceptable concentration and purity ratios were sent for Sanger sequencing. The Sanger sequencing data were reviewed using the Chromas software (version 2.6.6) (C. McCarthy; accessed on 20 January 2022), http://www.technelysium.com.au/chromas2.html).

### 2.4. RNA Sequencing

Total RNA from the whole blood was isolated using QIAamp^®^ RNA Blood Mini and processed according to the manufacturer’s instructions. The concentration and purity of the extracted RNA was assessed using the Nanodrop ND-1000 UV-VIS Spectrophotometer. RNA integrity numbers (RIN) were determined using an Agilent RNA 6000 Nano Kit with the Agilent 2000 Bioanalyser. Only samples with RIN > 7 were selected for RNA sequencing. Messenger RNA (mRNA) library preparation was done using Agilent’s SureSelect Strand-Specific RNA library preparation for Illumina Multiplex Sequencing. The steps for library preparation were as follows: (1) poly(A) RNA purification from total RNA; (2) first-strand cDNA synthesis and purification using AMPure XP beads; (3) second-strand cDNA synthesis, end repair and purification using AMPure XP beads; (4) dA-Tail of the cDNA 3′ ends; (5) ligation of the adapters, followed by their purification using AMPure XP beads; (6) amplification and indexing of the adapter-ligated cDNA library and purification with AMPure XP beads; (7) quality assessment using DNA 1000 chip (Bioanalyser 2100) and ScreenTape (using Tapestation system). The library concentration was visualised using an electropherogram, which displayed a single peak with a size between 200–600 bp.

All library preparations were sent to Theragen Etex Inc. (Seoul, Korea) for transcriptome sequencing, which was done using Illumina’s NovaSeq 6000 (150 bp paired-end reads; 200 million reads per sample). Results were received in a raw format and processed in-house. The following Illumina adapter sequences were used in the library preparation: read 1 (AGATCGGAAGAGCACACGTCTGAACTCCAGTCA) and read 2 (AGATCGGAAGAGCGTCGTGTAGGGAAAGAGTGT). The quality of the sequencing raw data was inspected using FastQC (version 0.11.8). Low-quality reads (reads with Phred scores less than Q20), adapter and poly G sequences were removed using Fastp (Appendix A: RNA sequencing reads and alignments, Appendix A: RNA Sequencing FASTP summary). The cleaned reads were referenced against Ensembl’s Human Reference Genome (version 38, Ensembl, Cambridge, United Kingdom) using HISAT2 (version 2.1.0). Aligned reads were quantified using featureCounts (version 1.6). Differential gene expression analyses were performed using DESeq2. Raw read and internal normalisation were performed across all samples, followed by customised sample-level and gene-level quality controls. Principal component analysis (PCA) was conducted to exclude potential outliers that could affect downstream analyses. Next, DEGs were filtered based on the following criteria: baseMean > 100, *p*-adjusted value (padj) < 0.01 and log2 fold change (lfc) > 1 and <−1. DEG profiles for genes with variants discovered in the ArcherDx targeted DNA sequencing panel were selected, and expression profiles were inspected during disease presentation and after attaining CR1/CR2. A heatmap to represent the DEG of the genes with variants was generated with hierarchical clustering based on a Euclidean distance metric.

### 2.5. Functional and Pathway Enrichment Analysis of Genes with Somatic Variants Using WEB-Based Gene Set Analysis Toolkit (WebGestalt) for Overrepresentation Analysis (ORA)

WebGestalt was utilised to perform gene ORA on genes present with somatic variants that were deregulated in this cohort [47]. ORA analysis was conducted to determine if the a priori gene set with variants in this study was present more than would be expected by chance [48]. A total of 26 genes with somatic variants were included in the ORA analysis for pathway and gene ontology analysis using WebGestalt. Customised filtering parameters were applied for ORA analysis as follows: (1) a minimum of three genes for a category, (2) Benjamini–Hochberg multiple test adjustment was chosen and (3) categories were first ranked based on their FDR, and then the most significant categories were selected for significance-based filtering. The Kyoto Encyclopaedia of Genes and Genomes (KEGG) database was utilised to identify affected pathways based on a *p*-value < 0.05 and a false discovery rate of ≤0.05. Based on the filtering criteria, seven out of ten pathways were inferred as significant, as summarised in Appendix A [49].

Next, gene ontology for biological processes (BP), cellular components (CC) and molecular functions (MF) was conducted using WebGestalt for the 26 genes present with somatic variants. Customised filtering parameters were applied for ORA analysis as follows: (1) a minimum of three genes for a category, (2) Benjamini–Hochberg multiple test adjustment was chosen and (3) categories were first ranked based on their FDR, and then the most significant categories were selected for significance-based filtering based on *p*-value <0.05 and a false discovery rate of ≤0.05. A total of 10 BP categories were significantly enriched, as summarised in Appendix A. As for the MF, 1/10 and 3/10 for the CC categories met the filtering criteria (Appendix A).

### 2.6. Statistical Analysis

Descriptive statistics were generated using IBM SPSS Statistics 22 software (SPSS, Chicago, IL, USA). The gene mutations and gene expression profiles were assessed using the Fisher exact test, and a *p*-value of less than 0.05 was considered significant to scrutinise if there was concordance between the mutational and gene expression profiles of these genes.

## 3. Results

### 3.1. Demographic and Clinical Summary of Patients in This Study

A total of eight de novo AML-NK patients whose DNA and RNA were collected at presentation and after CR1/CR2 were retrospectively included in this study. The median age of diagnosis was 40.5 years (range: 21–55 years). Six out of eight patients were female. All patients were negative for 30 fusion genes tested using a Leukaemia Q-Fusion Screening Kit (QuanDx, San Jose, CA, USA). The genotype for the *FLT3-ITD* and *NPM1* mutations revealed that four patients’ genotypes were *FLT3-ITD*^+^/*NPM1*^+^, two were *NPM1*^+^*FLT3*^wt^, and two were *NPM1*^wt^ *FLT3*^wt^. Aberrant antigen expressions were seen in four cases (three CD2^positive^ and one CD56^positive^).

The remission status of patients at CR1 was assessed based on post-induction chemotherapy that included a combination of cytarabine and anthracycline (daunorubicin) (DA 3+7) and consolidation chemotherapy that included HIDAC, MIDAC or FLAG. The patient’s remission at CR2 was assessed after salvage therapy using FLAG-IDA. MRD monitoring was based on either negativity for genetic markers (*NPM1* and/or *FLT3-ITD*) or multiparametric flow cytometry for cases without genetic markers [6]. Based on the ELN 2022 risk classification by genetics, 50% (4/8) of the patients had intermediate prognoses, and the others had good prognoses. Patients (DX1-7) were given the standard induction chemotherapy of daunorubicin and cytarabine (DA 3+7) followed by HIDAC/MIDAC/FLAG as the consolidation for patients who attained CR1. Patient DX8 was given salvage chemotherapy with FLAG-IDA, and they attained CR2. Four patients underwent stem cell transplantation, and all of the patients had an overall survival of more than five years in this study. The patients’ demographic, clinical and laboratory details are listed in Table 1.

### 3.2. Variants Discovered in This Study

An initial total of 208 variants (122 in the presentation and 86 in the CR1/CR2) were detected across the eight AML-NK samples. Rigorous filtering criteria were applied to exclude variants with MAF ≥ 1% and potential sequencing artefacts. Sanger sequencing was performed on the ASXL1 c.1934dup G646WfsTer12 variant recurrently seen in 12/16 samples to confirm if variants with VAF < 5% and/or that were recurrent in most samples were true findings. Sanger sequencing confirmed that this finding was a sequencing artefact. Hence for subsequent analyses, only cases with VAF > 5% were included to avoid the false positive inclusion of variants. After filtering, 131 recurrent (23/42) and non-recurrent (23/42) somatic variants involving 26 genes were identified, and they are summarised in Appendix A. Of these 42 variants, 21 missense, seven frameshifts, one splice donor variant, two in-frame insertions, and 11 untranslated regions (UTR)/intronic variants were affirmed.

The oncogenic effects and role of the somatic variants were assessed in various databases (dbSNP, Clinvar and COSMIC) and published in the literature. Novel putative somatic variants were assessed using computational variant effect prediction tools. Based on ACMG and AMP classification, the pathogenicity of the somatic variants was determined as follows: 18/42 (42.9%) as pathogenic, 4/42 (9.5%) as likely pathogenic, 4/42 (9.5%) as variant of known significance (VUS), 7/42 (16.7%) as likely benign and 9/42 (21.4%) as benign. Overall, the types of somatic variants that were detected were SNVs (26/42,61.9%), insertions (11/42, 26.2%) and deletions (5/42, 11.9%). The pathogenic variants were mainly SNVs (12/22, 54.5%) and insertions (8/22, 36.4%), and the rest were deletions (2/22, 9.1%). A total of nine novel somatic variants were discovered in this study (three likely pathogenic, one VUS and four benign/likely benign (Appendix A). Three likely pathogenic novel somatic variants were detected in the *CEBPA* gene (DX5 and DX8) involving regions that are within previously reported pathogenic variants in the COSMIC database [38].

### 3.3. Size of FLT3 Internal Tandem Duplication (ITD)

We detected *FLT3-ITD*s in four patients in this cohort (DX1, DX4, DX6 and DX7), as summarised in Table 1. The length of the ITDs ranged between 42–144 bp, with a median ITD length of 49.5 bp. The longest ITD was selected for patients with more than one ITD.

### 3.4. Evaluation of Somatic Variants Detected at Presentation and CR1/CR2

The samples in this cohort appeared to harbour between seven to thirteen somatic variants per sample; the minimum number of pathogenic somatic variants per sample was two, and the maximum was six. The mutational spectra of all somatic variants and their corresponding functional groups at disease presentation and after CR1/CR2 are depicted in Figure 1. Most recurrent somatic variants were detected belonging to cell signalling (*FLT3*), transcription (*RUNX1* and *CEBPA*), methylation (*DNMT3A*) and nucleoplasmin (*NPM1*) functional groups.

A total of 28 pathogenic/likely pathogenic variants (15 non-recurrent and 7 recurrent) were detected during disease presentation in all patients. Five somatic pathogenic/likely pathogenic variants were persistent in the CR1/CR2 samples (*DNTM3A* p.Arg659His, IDH2 p.Arg140Gln, LUC7L2 p.Glu253ArgfsTer34, *NOTCH1* p.Ala1740Val and *RUNX1* p.Leu56Ser genes). About 82% (23/28) of pathogenic/likely pathogenic variants seen during the presentation were lost after CR1/CR2. The VAF of pathogenic/likely pathogenic variants detected during the presentation and after CR1/CR2 in this study are depicted in Figure 2.

A total of twelve clonal haematopoiesis driver mutations were seen in eight genes, according to the 5th edition of the World Health Organization Classification of Haematolymphoid Tumours. The myeloid and histiocytic/dendritic neoplasms that were detected in this study that contribute to the increased risk of developing AML, as reported in COSMIC and published in the literature, are listed in Appendix A [12,38]. Of the twelve CH driver mutations, the persistently detected mutations in CR1/CR2 samples were *DNMT3A* p.Arg659His in patient DX1, *IDH2* p.Arg140Gln in patient DX3 and *NOTCH1* p.Ala1740Val in patient DX5.

The mutated genes were then classified into mutational classes (Class I, II and III) based on their putative effects that led to leukaemic transformation based on established leukaemogenesis models. Class I mutations confer proliferation and survival, such as activation of signal transduction pathways, whereas Class II gene aberrations affect differentiation and apoptosis, and Class III gene alterations promote epigenetic modifications. A total of 13/42 somatic pathogenic variants were classified into Class I (*n* = 1, gene: *FLT3*), Class II (*n* = 7, gene: *CEBPA*, *NPM1*, *WT1*) and Class III (*n* = 3, gene: *DNMT3A*, *IDH2*, *TET2*) as depicted in Figure 3. All of the patients had at least one mutation in one of these classes; five patients had mutations in Class II and Class III genes, one patient had mutations in all three gene classes and two patients had mutations in the Class II gene. Only two mutations in Class III, those in DNMT3A c.1976G > A and IDH2 c.419G > A that encode epigenetic modifiers, were retained, with VAFs of 41.68% and 44.83%, respectively, at disease presentation and 9.55% and 30.4%, respectively, after CR1. Somatic variants in recurrently mutated genes in AML (*FLT3*, *NPM1*, *CEBPA* and *IDH2*) were detected in all patients implicated as classic AML drivers and/or CH driver and hotspot mutations. Two VUS were recurrently detected in three patients (*RUNX1* NM_001754.4:c.1270T > C) and two patients (*ZRSR2* NM_005089.3:c.283G > A) that need further investigation to elucidate their roles in the leukaemogenesis of AML.

Differential gene expressions for the 26 variants were determined from the RNA sequencing data to compare the expression profiles of patients during disease presentation and after attaining CR1/CR2 based on the normalisation and fold change information obtained from FeatureCounts (version 1.6) and DESeq2. Findings from the DEG profiles of patients during the presentation and after CR1/CR2 are summarised in Table 1 and depicted in a heatmap (Figure 4). The relationship between the presence of mutations and gene expression was investigated. We observed no significant relationship between mutations and gene expression in all genes except in the biallelic mutations in the *CEBPA* genes detected in patients DX5 and DX8. Although upregulation of the *CEBPA* gene was observed in other patients during the presentation, in cases with biallelic CEBPA gene mutations (DX5 and DX8), the fold changes were almost three times higher than in other cases with overexpression of the *CEBPA* gene. *RUNX1* gene mutations were detected in patients DX1-2 and DX7-8, and overexpression was detected in patients DX1, DX3-4 and DX6-7, indicating no concordance between the mutation and gene expression profiles of this gene. Upregulation of the *FLT3*, *WT1* and *NPM1* genes was seen in all presentation samples, though a heterogeneous level of expression was observed across disease presentation and CR1/CR2. In terms of overall DEG expression (presentation samples versus CR1/CR2 samples), log2-fold changes of 5.07, 7.06 and 1.37 (Padj 0.01 for both) were observed in *FLT3*, *WT1* and *NPM1* genes, respectively. On the other hand, downregulation of genes was observed among the presentation samples compared to the CR1/CR2 samples in *CBL*, *NOTCH1*, *PPM1D*, *RAD21*, *STAG2* and *TET2*. The normalised data for the presentation and CR1/CR2 samples, fold change and padj values generated from the DESeq2 are summarised in Appendix A.

### 3.5. CEBPA Mutation and Gene Expression Profile

Mutations in the *CEBPA* gene (known as transcription factor CCAAT/enhancer binding protein α) were detected in the frequently reported two prototypical classes of mutations: N-terminal mutations and C-terminal basic leucine zipper (bZIP) regions. Patients DX5 and DX8 had double (bi-allelic) mutations in both of these regions, as depicted in Figure 5. Further evaluation of the DX5 and DX8 patients’ *CEBPA*-specific gene expression profiles revealed significant upregulations in both patients (Figure 4). Fisher’s exact test statistically confirmed a significant association between the biallelic *CEBPA* mutation and its expression (*p* = 0.036).

### 3.6. Functional and Pathway Enrichment Analysis of 26 Genes with Somatic Variants Using WebGestalt ORA Analysis

To elucidate the relevance of all of the genes present with somatic variants, over-representation analysis (ORA) using the WebGestalt tool for GO and KEGG pathway enrichment. Enriched KEGG pathways are summarised in Appendix A. Pathway analysis of the KEGG pathway is depicted in Figure 6 (transcriptional misregulation and pathways in cancer). Other significantly enriched pathways are appended in Appendix A documents (Appendix A for the acute myeloid leukaemia pathway, Appendix A for central carbon metabolism in cancer, Appendix A for pathways in cancer and Appendix A for the RAS signalling pathway). Gene ontology depicting the genes’ biological processes, cellular components and molecular functions are available in Appendix A. All significantly enriched GO terms in BP, MF and CC are summarised in Appendix A. All of the enriched GO terms in BP, MF and CC are also illustrated using volcano plots in Appendix A.

## 4. Discussion

The genetic heterogeneity of AML could explain the varying responses of patients to the treatment regime, necessitating the need for tailored therapy based on the patient’s genomic findings. However, the challenges lie in identifying the oncogenic effect of specific genetic aberrations and their role in the progression of leukaemogenesis. With the advent of NGS technology and bioinformatics tools, more sensitive assessments of genomic aberrations and MRD genomic markers are available. Recent evaluations of genomic MRD markers and their associations with patient outcomes and survival were explored and reported in multiple studies [27].

This study compared the mutations detected in eight matched presentation and CR1/CR2 samples using a high-throughput 75-gene panel by Archer HGC VariantPlex Myeloid. We performed paired-end deep sequencing on the samples collected at disease presentation (a minimum of 7 million reads) and increased the sequencing depth to 10X (a minimum of 30 million reads) to ensure mutations with low VAFs and minor leukaemic cell clones were not missed in the CR1/CR2 samples. As suggested by Yan et al. (2021), Sanger sequencing was performed on the selected variant (*ASXL1* c.1934dup G646WfsTer12) with VAF < 5% [34]. Sanger sequencing disconfirmed this variant as a sequencing artefact; hence, for all of the subsequent analyses, only variants with VAF > 5% were included in this study. We disagree with Wang et al. (2020), as they have included all variants with VAF > 1%, suggesting that there are no standardised cut-offs [16]. We affirm that the laboratory’s internal validation can determine cut-offs for VAFs by performing Sanger sequencing for validating low VAFs in samples as performed in this study. We also added another facet by performing high throughput transcriptomic deep sequencing (200 million reads, PE, 150 bp) for the same cohort of patients during disease presentation and CR1/CR2. We then compared the gene mutations and DEG profiles of the genes to discover the impact of pathogenic mutations on gene expression. We found that cases with biallelic *CEBPA* mutations were significantly associated with its upregulation in two patients (DX5 and DX8). Nevertheless, no significant association was observed between mutation and expression profiles in other genes investigated in this cohort. Next, gene enrichment was assessed via ORA using the WebGestalt tool to investigate the pathways that are affected among KEGG pathways. As an outcome, five pathways were significantly enriched, affecting several hub genes: *FLT3*, *AML1* (also known as *RUNX1*) and HRAS.

This cohort consisted of AML-NK patients below 60 years of age with good prognoses (*n* = 4) and intermediate prognoses (*n* = 4) according to the ELN 2022 classification. All the patients were treated with the same treatment protocol during induction therapy (DA 3+7), including personalised HIDAC/MIDAC/FLAG for consolidation/salvage therapy, and they attained CR (n(CR1) = 7; n(CR2) = 1). Four patients had undergone SCT: n(allo-SCT = 3); n(auto-SCT = 1). All patients had an excellent OS of above five years with no indication of relapse (Table 1).

The number of pathogenic somatic variants detected in the patients ranged between two and six per patient with no association with age, unlike previous reports by other studies that disclosed the number of mutations was lower in younger patients (18 to 39 years old) than in those above 40 years of age [16,50]. This study’s findings concur with another local study that performed whole-exome sequencing (WES) and reported fewer somatic variants in their AML-NK cohort (*n* = 6) [10,23,28]. Although they performed WES sequencing with greater genome coverage, the number of somatic variants was relatively low, especially in the myeloid-related genes included in our targeted NGS panel. The number of somatic mutations detected in our cohort was lower than in other studies with similar targeted myeloid panel designs [16,51].

In this study, the overall ratio of indels to SNVs was 1.58 at presentation and 2.4 at CR1/CR2, whereas the ratio of pathogenic somatic indels to SNVs was 1.15 at presentation and 0.25 at CR1/CR2, in contrast to previous publications [52,53]. The higher overall ratios of indels to SNVs in our study are due to recurrent benign indels seen in *FLT3*, *RAD21* and *EZH2* genes, as illustrated in Appendix A. Our findings did not concur with another WES study conducted on AML-NK in another local study that exhibited a lower ratio of indels to SNVs, with SNVs being almost twice as common as indels [53]. The findings could be due to differences in experimental design, such as WES versus the 75-myeloid gene panel utilised in this study.

The variants detected in this study are depicted in Figure 1 based on their types and functional groups. This study saw mutations in the signalling and kinase pathway-related genes that caused abnormal activation and proliferation of cellular signalling pathways, also known as Class 1 mutations, seen in 6/8 cases in this study, which agrees with other studies [28,54]. Similar to other studies, we also identified mutations in the epigenetic modifiers that include regulation of DNA methylation and chromatin modification in 6/8 cases in this cohort, which are recognised as drivers in leukaemogenesis. Mutations in epigenetic modifiers result in clonal outgrowth but require subsequent mutations to initiate leukaemic transformation [28,54].

Mutations in the myeloid transcription factors were seen in 3/8 cases, which concords with other studies that reported their presence in approximately 20% to 25% of AML cases [54]. *CEBPA* is an essential transcription factor for haematopoietic lineage-specific myeloid differentiation. Studies have reported that CEBPA gene mutations were found in 10–15% of de novo AML cases, especially in AML-NK patients [55,56]. Typically, *CEBPA* mutations cluster in two focal hotspots: N-terminal frameshifts insertions/deletions and/or C-terminal in-frame insertions/deletions. Mutations in the N-terminal give rise to the truncated p30 protein, which presides negatively to other full-length p42 proteins, whereas mutations in the C-terminal will obstruct the binding of CEBPA to DNA or dimerisation. In this cohort, we found biallelic *CEBPA* mutations in two cases (patients DX5 and DX8) at the hotspot regions of the N-terminal (within the first 357 bp of coding sequence) and the C-terminal bZIP region (between c.834–1074) which likely yielded the truncated p30 protein and led to the disruption of the binding of CEBPA to DNA or to dimerization [57]. Upregulated *CEBPA* genes were observed distinctly in cases with biallelic *CEBPA* mutations, as discovered in other studies [58,59]. AML with a *CEBPA* mutation is still retained as a unique entity in the 2022 WHO classification, and patients with biallelic *CEBPA* mutations often have good prognoses [7,60]. *CEBPA* gene upregulation and the outcomes of AML patients were inconclusive in several studies [14,61]. Although ELN 2022 suggested in-frame mutations affecting the bZIP region of CEBPA, irrespective of whether monoallelic or biallelic occurrence is the favourable prognosis, we could not ascertain whether biallelic mutations accompanied with the upregulation of the *CEBPA* gene is a favourable prognostic factor because the DX5 patient attained CR1 following induction and had OS > 5 years without SCT, whereas the DX8 patient only attained CR2 and had undergone allo-SCT in this study.

The length of internal tandem duplication in the FLT*3-ITD* cases in this cohort was above the cut-offs (>30) used in several studies for prognosis based on the ITD’s length [62,63,64,65]. There are contradictory findings about the size of *FLT3-ITD* insertion length and the patient’s clinical outcome. Several studies have shown poor OS [65,66,67] with increasing ITD length, whereas increased ITD insertion conferred better OS in some studies [62,68]. However, some studies did not agree with either of these findings and indicated that the ITD length has no significance in the prognosis or clinical application of AML [69,70,71]. In this study, all patients had a good OS of above five years with concomitant *NPM1* mutation; hence, their prognoses were categorised as intermediate based on the ELN 2022 classification. In addition, we did not observe any relationship between the presence of *FLT3-ITD* and the ITD’s length with the expression of *FLT3* genes, as *FLT3* was overexpressed in all cases in this cohort. Moreover, the sample size was insufficient to infer the relationship between ITD insertion length and patient prognosis in this study.

Of the five pathogenic/likely pathogenic somatic variants that were persistent in the CR1/CR2 samples, three variants (*DNMT3A* p.Arg659His, *RUNX1* p.Leu56Ser in Patient DX1 and *NOTCH1* p.Ala1740Val in Patient DX5) were reported as having a germline predisposition in AML [37,72,73,74]. Two other recurrent pathogenic somatic variants that persisted after CR1/CR2, variants in IDH2 p.Arg140Gln (seen in Patients DX3 and DX6) and *LUC7L2* p.Glu253ArgfsTer34 (seen in Patients DX2, DX3 and DX8), were reported as somatic in the Clinvar database and in other publications [37,72,75,76]. We deduced that the pathogenic somatic variant detected in *NPM1* p.Trp288CysfsTer12, which was recurrent in six cases (Patients DX1, DX2, DX3, DX4, DX6, DX7), is suitable for MRD monitoring for treatment response in this cohort where the VAF at presentation ranged between 25–29% and was not detectable at CR1/CR2. We also observed interesting findings in the DEG profiles of FLT3, WT1 and NPM1 that exhibited upregulation and those of CBL, NOTCH1, PPM1D, RAD21, STAG2 and TET2 that were downregulated at presentation versus the CR1/CR2 patients. Studies revealed the utility of FLT3, WT1 and NPM1 expression for MRD monitoring in AML, which supports the findings in this study [77,78,79].

Functional enrichment analysis using ORA (WebGestalt) revealed that five pathways were significantly enriched (*p*-value < 0.01, FDR < 0.05), as illustrated in Appendix A. In addition, several hub genes were identified in several pathways, such as *FLT3*, *RUNX1* and *HRAS* (Appendix A). Transcription misregulation in cancer tops the list of affected pathways, as some of the upstream genes in the transcriptional regulation, such as *CEBPA* and *RUNX1*, were deregulated in most cases, as shown in Figure 6. The overexpression of *FLT3* impacted JAK-STAT signalling and cytokine–cytokine receptor interaction, as this gene is also located upstream of the pathway (Appendix A). Overexpression of *RUNX1*, which is a hub-gene crucial in haematopoietic differentiation, was observed in 62.5% (5/8) of presentation samples (Figure 6). Overexpression of CEBPA genes was observed in 88% (7/8 of the presentation samples) of the patients with concurrent biallelic mutations (Patients DX5 and DX8), supporting the notion that haematopoietic resistance is evident in the pathogenesis of AML-NK (Figure 6). The variant in the *HRAS* gene detected in this study (Sample DX7) was benign, occurred in the 5′ UTR region, and did not affect gene expression in this study. However, two patients (DX1 and DX2) exhibited upregulation of this HRAS gene, as depicted in Figure 4. Upregulation of the *HRAS* gene could impact the deregulation of several pathways: PI3K-Akt signalling pathway, Jak-STAT signalling pathway, MAPK signalling pathway and calcium signalling pathway, as *HRAS* is a hub-gene in the RAS signalling pathway (Appendix A).

The most significantly enriched pathway in this study was transcriptional misregulation in cancer (hsa05202) (*p* < 0.01, FDR = 0.001), which was closely related to the most enriched GO MF category, DNA-binding transcription activator activity related to RNA polymerase II-specific (GO:0001228) (*p* < 0.01, FDR = 0.4), with four overlapping genes that includes hub genes such as *FLT3* and *CEBPA*, as listed in Appendix A. The ancestral GO:0140110 (Transcription regulation activity) for GO:001228 (Appendix A) ensures transcriptional regulators are modulating the gene expression at the right cells at the right time, which were disrupted in this study as the upstream genes (*AML1/RUNX1* and *CEBPA*) at the hsa05202 (Figure 6) were dysregulated, as depicted in the DEGs of genes during disease presentation and after CR1/CR2 in Figure 4. This explains the transcriptional dysregulation because the majority of the signalling pathways target transcription machinery, which provides insights into the mechanism of AML-NK leukaemogenesis [80].

### Significance of the Study

Primarily, this study classified mutations discovered in AML-NK patients based on various classification guidelines (ACMG/AMP), databases (dbSNP, Clinvar and COSMIC) and variant effect prediction tools. Based on the ACMG/AMP recommendation, the variants were classified as benign, likely benign, VUS, likely pathogenic and pathogenic. The variants were then assessed based on mutational classes (Class I, II and III) and their functional classes to explicate their putative effects that enabled leukaemic transformation based on established leukaemogenesis models. Then, the variants detected during disease presentation and after CR1/CR2 were assessed to identify potential MRD biomarkers for treatment monitoring. Next, gene expression profiles for 26 genes with somatic variants were determined to assess the effect of these variants on their DEG profiles. Next, DEG profiles for 26 genes with somatic variants were assessed during disease presentation and after CR1/CR2. Finally, ORA analysis was conducted using the WebGestalt tool for GO and KEGG pathway enrichment analyses of the 26 genes with their variants in this study.

To recapitulate, this study demonstrated the mutational profiles depicting AML’s genomic landscape heterogeneity. Each patient had a unique list of somatic pathogenic variants that belonged to Class I/II/III mutations. In some cases, CH-related variants were identified that conferred an increased risk of developing AML. The deregulation of genes with mutations impacted the pathways in cancer and various signalling pathways, as these genes are either hub genes or were upstream of pathways, as described earlier. Based on ELN 2022 classifications, the two patients (DX5 and DX8) with biallelic CEBPA gene mutations affecting the bZIP regions were assigned to a favourable prognosis group (Table 1). We also verified the *NPM1* p.Trp288CysfsTer12 mutation as a recurrent biomarker that conferred a favourable prognosis in our cohort. In addition, we also exemplified the usefulness of the DEG profiles of *FLT3*, *WT1* and *NPM1* for MRD assessment in AML-NK patients.

## 5. Conclusions

This study elucidated the genomic mutations and gene expression profiles of myeloid genes using a 75-gene targeted DNA panel (Archer HGC VariantPlex Myeloid) during disease presentation and after CR1/CR2. Diverse mutations and expression profiles were observed in each patient during disease presentation, indicating that AML-NK is indeed a heterogeneous disorder requiring further risk-based stratification. This study discovered three novel pathogenic somatic variants at the reported prototypical regions, N-terminal and bZIP regions of the *CEBPA* gene that were significantly associated with its upregulation. Although the cohort size is relatively small, we identified potential biomarkers that indicate favourable prognoses and are suitable for MRD monitoring, including *NPM1* p.Trp288CysfsTer12, which was recurrent in 75% (6/8) of the patients and is in line with the ELN 2022 recommendation. We also suggested DEG profiles for *FLT3*, *WT1*, and *NPM1* for MRD as potential biomarkers for MRD assessment in AML-NK patients. This multifaceted study comprehensively integrated DNA and RNA sequencing findings with functional enrichment in AML-NK that has not been described elsewhere. Our next step is to expand this study into a larger cohort of AML-NK patients to formulate a hierarchical prognosis model with unique mutations and gene expression profiles. 

## Figures and Tables

**Figure 1 cancers-15-01386-f001:**
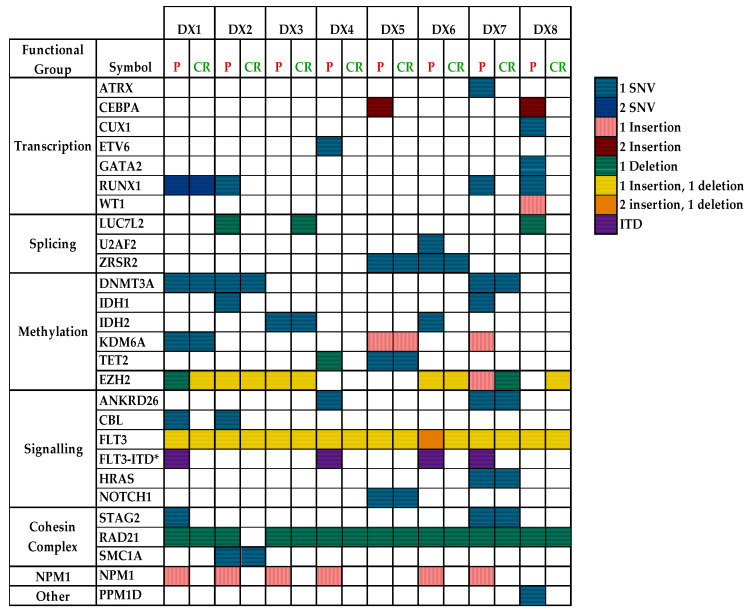
The mutational spectra of all somatic variants (pathogenic/benign/VUS) and their corresponding functional groups at disease presentation and after CR1/CR2. The coloured boxes indicate the types of variants detected in the study SNV/insertion/deletion/ITD as depicted in the legend. P refers to samples collected at presentation, and CR refers to samples collected after remission (CR1/CR2). * Refers to the internal tandem duplication (ITD).

**Figure 2 cancers-15-01386-f002:**
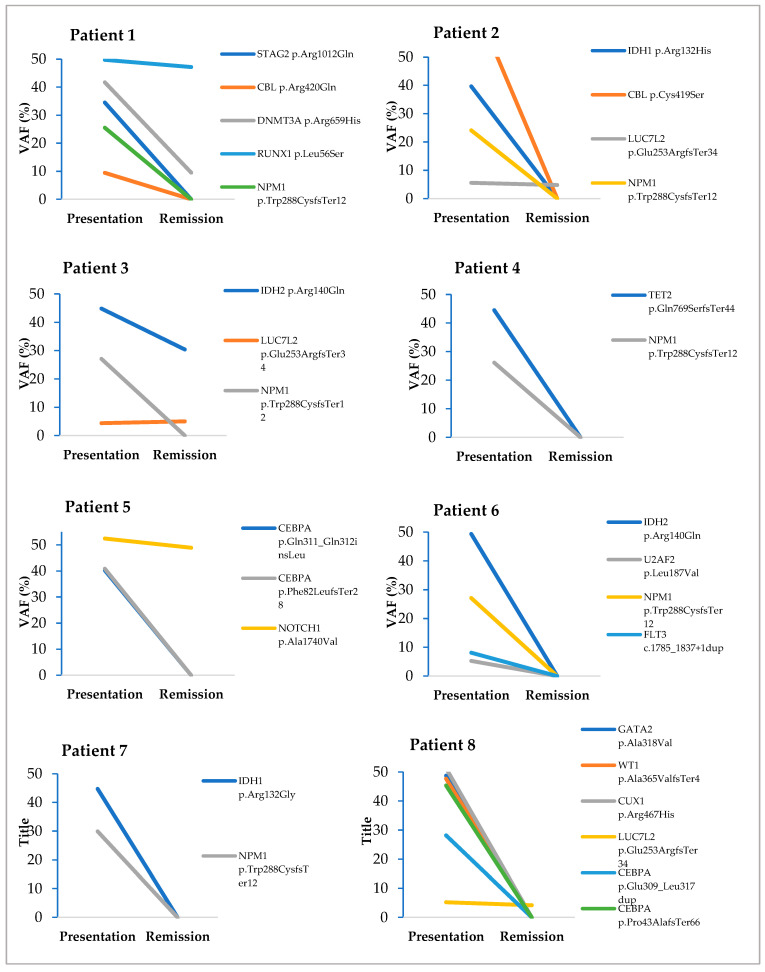
Comparison between VAFs during disease presentation and after CR1/CR2. Each line in the graph corresponds to a somatic pathogenic variant detected during disease presentation and after CR1/CR2. Only pathogenic variants were included in each patient’s VAF comparison during the presentation and after CR1/CR2.

**Figure 3 cancers-15-01386-f003:**
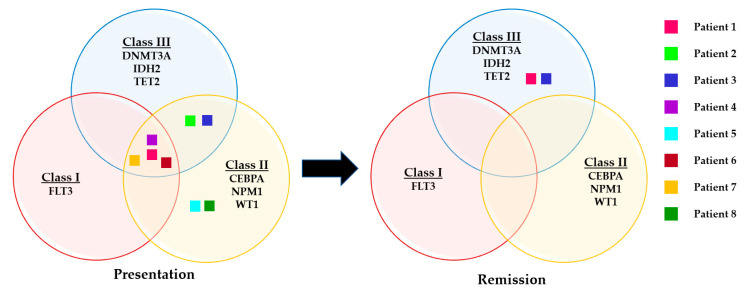
Representation of somatic variants during disease presentation and after attainment of CR1/CR2 based on the mutational classes.

**Figure 4 cancers-15-01386-f004:**
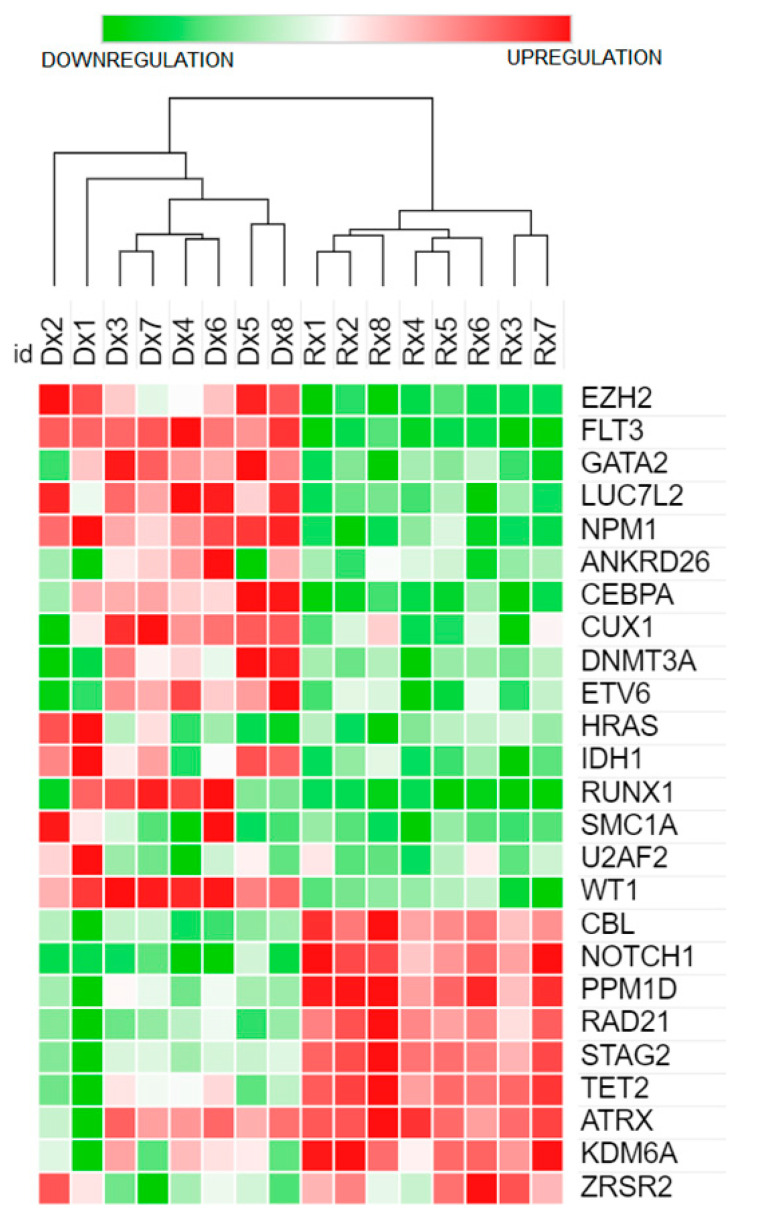
Heatmap of genes with variants during disease presentation and after attainment of CR1/CR2.

**Figure 5 cancers-15-01386-f005:**
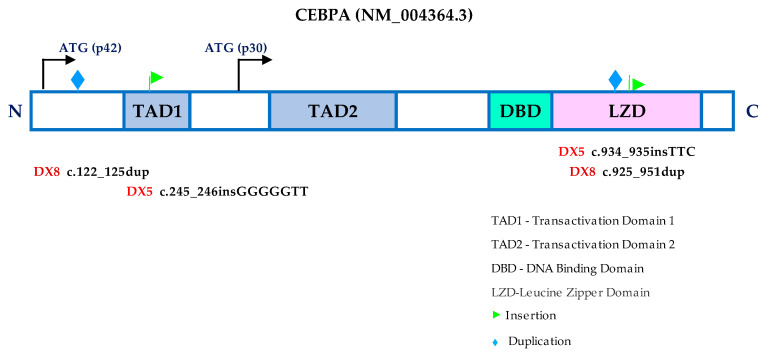
A schematic representation of the *CEPBA* gene structure with two alternative start sites that give rise to different isoforms (p42 and p30). Patients DX5 and DX8 had biallelic frameshift mutations at the N-terminal region and in-frame insertion at the C-terminal bZIP mutations.

**Figure 6 cancers-15-01386-f006:**
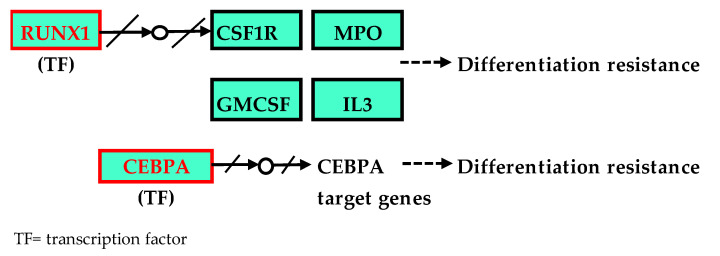
Transcriptional misregulation pathway and pathways in cancer (reference: KEGG pathways) [12,48]. This pathway illustrates the affected hub-gene *RUNX1* and the upstream gene, CEBPA, in red boxes, which elucidates the differentiation resistance in AML-NK patients in this cohort in cases with deregulated *CEBPA* and *RUNX1* genes. Upregulation of the *RUNX1* and *CEBPA* genes could lead to impaired resistance to normal haematopoietic differentiation.

**Table 1 cancers-15-01386-t001:** Demographic, clinical and laboratory information of patients in this cohort.

				Patient Details				
	DX1	DX2	DX3	DX4	DX5	DX6	DX7	DX8
**Demographic data**								
Age	31	29	54	55	36	49	45	21
Ethnicity	Malay	Malay	Indian	Malay	Chinese	Malay	Indian	Malay
Gender	Female	Female	Female	Female	Female	Female	Male	Male
No. of mutations	3	3	1	1	1	3	1	5
**Karyotype**								
G-banding	46 (X,X) [19]	46 (X,X) [19]	46 (X,X) [19]	46 (X,X) [19]	46 (X,X) [19]	46 (X,X) [19]	46 (X,Y) [19]	46 (X,Y) [19]
**Cell counts**								
White blood cell count (109/L)	26.9	57.7	31.1	170	19.5	45.8	39.5	152.8
Haemoglobin (g/dL)	7.3	8.1	8.1	8.4	11.3	7.9	9.9	6.1
Platelet (109/L)	32	76	83	70	68	21	22	27
% Blast (PB)	20%	55%	89%	90%	40%	90%	95%	94%
% Blast (BMA)	90%	80%	90%	92%	90%	96%	90%	95%
**Flow cytometry immunophenotyping**								
Aberrant antigen expression	CD2+	CD2+	-	CD56+	CD2+	-	-	-
**Gene mutations**								
Leukaemia Q-Fusion (30 fusion genes)	Negative	Negative	Negative	Negative	Negative	Negative	Negative	Negative
FLT3-ITD	Detected	wt	wt	Detected	wt	Detected	Detected	wt
NPM1 mutation	Detected	Detected	Detected	Detected	wt	Detected	Detected	wt
**Archer HGC VariantPlex Myeloid Panel**								
Genes with pathogenic/likely pathogenic somatic variants/ITD	CBL	CBL	IDH2	TET2	CEBPA	FLT3	IDH1	CEBPA
	DNMT3A	IDH1	NPM1	NPM1		IDH2	NPM1	CUX1
	RUNX1	LUC7L2		FLT3-ITD		U2AF2	FLT3-ITD	GATA2
	STAG2	NPM1				NPM1		IDH1
	NPM1					FLT3-ITD		LUC7L2
	FLT3-ITD							WT1
FLT3-ITD insertion length (Archer HGC)	42	-	-	45	-	54	144	-
Number of mutations per patients	4	3	1	1	1	3	1	5
**DEG**								
Upregulated genes	EZH2	EZH2	CUX1	CUX1	EZH2	ANKRD26	CUX1	EZH2
	FLT3	FLT3	FLT3	ETV6	CEBPA	CUX1	FLT3	CEBPA
	HRAS	HRAS	GATA2	FLT3	CUX1	FLT3	GATA2	CUX1
	IDH1	LUC7L2	LUC7L2	GATA2	DNMT3A	GATA2	NPM1	DNMT3A
	NPM1	NPM1	NPM1	LUC7L2	FLT3	LUC7L2	RUNX1	ETV6
	RUNX1	SMC1A	RUNX1	NPM1	GATA2	NPM1	WT1	FLT3
	U2AF2		WT1	RUNX1	IDH1	RUNX1		GATA2
				WT1	NPM1	SMC1A		IDH1
						WT1		LUC7L2
								NPM1
Downregulated genes	ATRX	ATRX	CBL	CBL	CBL	CBL	CBL	TET2
	CBL	CBL	NOTCH1	NOTCH1	NOTCH1	NOTCH1	KDM6A	STAG2
	KDM6A	KDM6A	PPM1D	PPM1D	PPM1D	PPM1D	NOTCH1	RAD21
	NOTCH1	NOTCH1	RAD21	RAD21	RAD21	RAD21	PPM1D	PPM1D
	PPM1D	PPM1D	STAG2	STAG2	STAG2	STAG2	RAD21	NOTCH1
	RAD21	RAD21	TET2	TET2	TET2	TET2	STAG2	KDM6A
	STAG2	STAG2					TET2	CBL
	TET2	TET2						
**ELN 2017 Classification**								
Prognosis	Intermediate	Good	Good	Intermediate	Good	Intermediate	Intermediate	Good
**Treatment Protocol**								
Induction	DA 3+7	DA 3+7	DA 3+7	DA 3+7	DA 3+7	DA 3+7	DA 3+7	DA 3+7
Consolidation	MIDAC/HIDAC	HIDAC/HIDAC/HIDAC	MIDAC/HIDAC/FLAG	MIDAC/FLAG/HIDAC	MIDAC/HIDAC/ARAC	MIDAC/FLAG/FLAG	MIDAC/HIDAC/ARAC	FLAG-IDA/HIDAC/ARAC
SCT	Nil	Nil	Nil	Allo-SCT after CR1	Nil	Auto-SCT after CR1	Allo-SCT after CR1	Allo-SCT after CR2
**Response to treatment**								
Remission status	CR1	CR1	CR1	CR1	CR1	CR1	CR1	CR2
OS	>5	>5	>5	>5	>5	>5	>5	>5

## Data Availability

The datasets generated and/or analysed during the current study are available in the Sequence Read Archive, National Center for Biotechnology Information [SRA, NCBI] repository, Accession: PRJNA902950. Other data are available in Appendix A.

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
