# Peer review of "Genomic Alterations, Gene Expression Profiles and Functional Enrichment of Normal-Karyotype Acute Myeloid Leukaemia Based on Targeted Next-Generation Sequencing"

_cancers, 2023, doi:10.3390/cancers15051386_

Round 1
Reviewer 1 Report
The authors investigated gene expression and gene alterations in eight AML patients with normal karyotype. The manuscript is excellent and exciting. Thank you.
Author Response
Dear Reviewer,
Thank you for the positive feedback and invaluable feedback. We truly appreciate it.
Reviewer 2 Report
Ambayya and colleagues apply a targeted NGS DNA panel and RNA-seq to 8 normal karyotype (NK) AML samples. The experiments are well planned and the bioinformatics analysis has been carried out correctly. The authors have exploited the data extremely as they have carried out a thorough analysis of every possible aspect.
English editing must be done. Focusing on scientific issues, the main limitation of the article is the short cohort (acknowledged by the authors throughout the text). By analyzing 8 patients it is impossible to draw conclusions regarding the prognosis (line 590). Furthermore, NK AML has been extensively studied in large cohorts, both at the DNA and RNA level, so the data shown is not novel. In fact, because they use a targeted myeloid panel, no new altered genes have been found. Another important aspect is that the latest ELN guidelines have not been cited or taken into account for molecular risk stratification
Author Response
Dear Reviewer (Replied here and also as attachment for your kind perusal)
Response to Reviewer 2 comments
Dear Reviewer,
Thank you very much for your invaluable feedback and comments.
In response to your review, we have made amendments as follows:
- English editing must be done.
Response: We have done thorough English editing as required by the journal
- By analyzing 8 patients it is impossible to draw conclusions regarding the prognosis (line 590).
Response: we have clarified the conclusion by including additional statements and evidence for the conclusion as highlighted in the main text as follows:
Although the cohort size is relatively small, we identified potential biomarkers that indicate a favourable prognosis and are suitable for MRD monitoring, including NPM1 p.Trp288CysfsTer12, which was recurrent in 75% (6/8) of the patients, in line with the ELN 2022 recommendation. We also suggested DEG profiles for FLT3, WT1, and NPM1 for MRD as potential biomarkers for MRD assessment in AML-NK patients. This multifaceted study comprehensively integrated DNA and RNA sequencing findings with functional enrichment in AML-NK that has not been described elsewhere. Our next step is to expand this study into a larger cohort of AML-NK patients to formulate a hierarchical prognostication model with unique mutations and gene expression profiles.
- Furthermore, NK AML has been extensively studied in large cohorts, both at the DNA and RNA level, so the data shown is not novel.
Response: We deny that the findings in this study are “not novel”. This multifaceted study comprehensively integrated DNA and RNA sequencing findings with functional enrichment in AML-NK that has not been described elsewhere. No other study has correlated the gene expression profiles with the patient’s mutation spectrum to our knowledge based on thorough and systematic literature reviews. The genes used in the targeted NGS panel are a hotspot for myeloid malignancies, and this is a preliminary study to assess the NGS panel before we move on to a more extensive whole genome sequencing-based study. The methodology described in this study is novel, especially for verifying the pathogenicity of variants using various variant effect prediction tools. We did not merely report previously published known mutations on these genes but verified new mutations not reported elsewhere based on methodologies described comprehensively in this study.
We would also like to reiterate that though the sample size was only eight patients, we followed up on these patients after CR1/CR2 and their remission samples were utilised to investigate the potentiality for a MRD study. It was very fortunate that although the sample size was small, we were able to retrieve the follow-up samples (CR1/CR2) and obtain survival data of more than five years in this cohort. The findings in this small cohort will benefit as we have identified recurrent biomarkers and confirmed findings as reported by WHO 2022 and ELN 2022, as highlighted throughout the manuscript. Additionally, we have performed functional enrichment analyses and elucidated the role of these genes in AML pathogenesis by describing the hub genes and upstream genes. We duly acknowledged the limitation of the small cohort, but it is interesting to consider the depth of our analysis and description of the cases analysed in this cohort.
- Another important aspect is that the latest ELN guidelines have not been cited or taken into account for molecular risk stratification.
Response: Sincere apologies for overlooking ELN 2022. We missed the latest ELN guidelines as this manuscript was initially written before the release of the guidelines. We have revamped the manuscript in regards to ELN 2022 guidelines as suggested (highlighted throughout the manuscript) and cited.
This manuscript aims to perform the genetic profiling of AML-NK patients during the patient’s presentation and after the attainment of CR1/CR2 by integrating DNA and RNA sequencing findings with functional enrichment analysis. Targeted DNA sequencing was utilised as we look forward to setting up these assays and pipelines for clinical diagnosis in our centre based on the availability of resources and finances. Ultimately, we aim to discover clinically relevant biomarkers that can be targeted with currently available precision medicine in AML-NK. In future studies, we will explore the whole genome of AML-NK to elucidate novel genes.
Response to Reviewer 2 comments
Dear Reviewer,
Thank you very much for your invaluable feedback and comments.
In response to your review, we have made amendments as follows:
- English editing must be done.
Response: We have done thorough English editing as required by the journal
- By analyzing 8 patients it is impossible to draw conclusions regarding the prognosis (line 590).
Response: we have clarified the conclusion by including additional statements and evidence for the conclusion as highlighted in the main text as follows:
Although the cohort size is relatively small, we identified potential biomarkers that indicate a favourable prognosis and are suitable for MRD monitoring, including NPM1 p.Trp288CysfsTer12, which was recurrent in 75% (6/8) of the patients, in line with the ELN 2022 recommendation. We also suggested DEG profiles for FLT3, WT1, and NPM1 for MRD as potential biomarkers for MRD assessment in AML-NK patients. This multifaceted study comprehensively integrated DNA and RNA sequencing findings with functional enrichment in AML-NK that has not been described elsewhere. Our next step is to expand this study into a larger cohort of AML-NK patients to formulate a hierarchical prognostication model with unique mutations and gene expression profiles.
- Furthermore, NK AML has been extensively studied in large cohorts, both at the DNA and RNA level, so the data shown is not novel.
Response: We deny that the findings in this study are “not novel”. This multifaceted study comprehensively integrated DNA and RNA sequencing findings with functional enrichment in AML-NK that has not been described elsewhere. No other study has correlated the gene expression profiles with the patient’s mutation spectrum to our knowledge based on thorough and systematic literature reviews. The genes used in the targeted NGS panel are a hotspot for myeloid malignancies, and this is a preliminary study to assess the NGS panel before we move on to a more extensive whole genome sequencing-based study. The methodology described in this study is novel, especially for verifying the pathogenicity of variants using various variant effect prediction tools. We did not merely report previously published known mutations on these genes but verified new mutations not reported elsewhere based on methodologies described comprehensively in this study.
We would also like to reiterate that though the sample size was only eight patients, we followed up on these patients after CR1/CR2 and their remission samples were utilised to investigate the potentiality for a MRD study. It was very fortunate that although the sample size was small, we were able to retrieve the follow-up samples (CR1/CR2) and obtain survival data of more than five years in this cohort. The findings in this small cohort will benefit as we have identified recurrent biomarkers and confirmed findings as reported by WHO 2022 and ELN 2022, as highlighted throughout the manuscript. Additionally, we have performed functional enrichment analyses and elucidated the role of these genes in AML pathogenesis by describing the hub genes and upstream genes. We duly acknowledged the limitation of the small cohort, but it is interesting to consider the depth of our analysis and description of the cases analysed in this cohort.
- Another important aspect is that the latest ELN guidelines have not been cited or taken into account for molecular risk stratification.
Response: Sincere apologies for overlooking ELN 2022. We missed the latest ELN guidelines as this manuscript was initially written before the release of the guidelines. We have revamped the manuscript in regards to ELN 2022 guidelines as suggested (highlighted throughout the manuscript) and cited.
This manuscript aims to perform the genetic profiling of AML-NK patients during the patient’s presentation and after the attainment of CR1/CR2 by integrating DNA and RNA sequencing findings with functional enrichment analysis. Targeted DNA sequencing was utilised as we look forward to setting up these assays and pipelines for clinical diagnosis in our centre based on the availability of resources and finances. Ultimately, we aim to discover clinically relevant biomarkers that can be targeted with currently available precision medicine in AML-NK. In future studies, we will explore the whole genome of AML-NK to elucidate novel genes.

Reviewer 3 Report
I found this to be an interesting manuscript. The authors explored the genomic aberrations in the acute myeloid leukemia normal karyotype genome in eight patients using a targeted NGS panel and RNA sequencing for gene expression profiling to reveal differentially expressed genes and they interpreted functional and pathway enrichment.
The authors identified genomic biomarkers that could be useful in disease monitoring for acute myeloid leukemia normal karyotype patients.
I found that the question posed by the authors was well-defined. The methods are appropriate and well-described. The data is sound and well-controlled. The manuscript adheres to the relevant standards for reporting and data deposition. The discussion and conclusions are well-balanced and adequately supported by the data. The writing was acceptable.
Author Response
Dear Reviewer,
Thank you for the positive and invaluable feedback. We truly appreciate it.
Round 2
Reviewer 2 Report
“This manuscript aims to perform the genetic profiling of AML-NK patients during the patient’s presentation and after the attainment of CR1/CR2 by integrating DNA and RNA sequencing findings with functional enrichment analysi […] “ We deny that the findings in this study are “not novel”. This multifaceted study comprehensively integrated DNA and RNA sequencing findings with functional enrichment in AML-NK that has not been described elsewhere. No other study has correlated the gene expression profiles with the patient’s mutation spectrum to our knowledge based on thorough and systematic literature reviews.”
“The methodology described in this study is novel, especially for verifying the pathogenicity of variants using various variant effect prediction tools. We did not merely report previously published known mutations on these genes but verified new mutations not reported elsewhere based on methodologies described comprehensively in this study”
We acknowledge that the exact same methods have not been applied elsewhere. However, similar approaches (combination of DNA and RNA sequencing approaches) in large series can be found in literature:
· Papaemmanuil et al. Genomic Classification and Prognosis in Acute Myeloid Leukemia. N Engl J Med. 2016 Jun 9;374(23):2209-2221.
· Functional genomic landscape of acute myeloid leukaemia. Tyner et al. Nature volume 562, pages526–531 (2018)+
· Ediriwickrema A, Aleshin A, Reiter JG, Corces MR, Köhnke T, Stafford M, Liedtke M, Medeiros BC, Majeti R. Single-cell mutational profiling enhances the clinical evaluation of AML MRD. Blood Adv. 2020 Mar 10;4(5):943-952.
· Docking et al. A clinical transcriptome approach to patient stratification and therapy selection in acute myeloid leukemia. Nat Commun 12, 2474 (2021).
· Petty et al Genetic and Transcriptional Contributions to Relapse in Normal Karyotype Acute Myeloid Leukemia. Blood Cancer Discov (2022) 3 (1): 32–49. RESEARCH ARTICLES| JANUARY 01 2022
· Murdock et al. Impact of diagnostic genetics on remission MRD and transplantation outcomes in older patients with AML, Blood,Volume 139, Issue 24, 2022, Pages 3546-3557,
· Tazi et al. Unified classification and risk-stratification in acute myeloid leukemia. Nat Commun 2022; 13:4622.
Thus, although the scientific rigor and the thorough analysis are impeccable, the conclusions highlighted by the authors in their response are broadly known:
“ we identified potential biomarkers that indicate a favourable prognosis and are suitable for MRD monitoring, including NPM1 p.Trp288CysfsTer12, which was recurrent in 75% (6/8) of the patients, in line with the ELN 2022 recommendation” à NPM1 favorable prognosis has been acknowledged since the first ELN recommendations for risk stratification. Moreover, some protocols have incorporated this genetic markers for MRD follow up based on previous sound research by collaborative groups:
· Falini, B., Sciabolacci, S., Falini, L. et al. Diagnostic and therapeutic pitfalls in NPM1-mutated AML: notes from the field. Leukemia 35, 3113–3126 (2021))
“We also suggested DEG profiles for FLT3, WT1, and NPM1 for MRD as potential biomarkers for MRD assessment in AML-NK patients” à this is the most relevant finding, but DEG is more difficult to apply and interpret than single qRT-PCR assays, which can be easily be applied in NPM1-mutated patients (in fact, commercial kits are available), or WT1 overexpressing, for who follow up with these genes has been suggested before
· Rossi G et al. Wilms’ Tumor Gene (WT1) Expression and Minimal Residual Disease in Acute Myeloid Leukemia. In: van den Heuvel-Eibrink MM, editor. Wilms Tumor [Internet]. Brisbane (AU): Codon Publications; 2016 Mar. Chapter 16.
· Liu H, Wang X, Zhang H, Wang J, Chen Y, Ma T, Shi J, Kang Y, Xi J, Wang M, Wang M, et al: Dynamic changes in the level of WT1 as an MRD marker to predict the therapeutic outcome of patients with AML with and without allogeneic stem cell transplantation. Mol Med Rep 20: 2426-2432, 2019
“ we have identified recurrent biomarkers and confirmed findings as reported by WHO 2022 and ELN 2022 “ à Indeed
“ is interesting to consider the depth of our analysis and description of the cases analysed in this cohort” à We agree on this
Author Response
Hi Reviewer 2,
Please refer to the attachment for detailed final response.
Thank you

Round 3
Reviewer 2 Report
As you mention your paper is interesting in terms of geographical location to explore genomics of AML-NK in Asia.
My general comment is all about an overall priority of the journal.